# Explaining Modern Gated-Linear RNNs via A Unified Implicit Attention Formulation

**Itamar Zimerman**[*]                **Ameen Ali**[*]                **Lior Wolf**

The Blavatnik School of Computer Science, Tel Aviv University
{zimerman1,ameenali}@mail.tau.ac.il, wolf@cs.tau.ac.il

## Abstract

Recent advances in efficient sequence modeling have led to attention-free layers, such as Mamba, RWKV, and various gated RNNs, all featuring sub-quadratic complexity in sequence length and excellent scaling properties, enabling the construction of a new type of foundation models. In this paper, we present a unified view of these models, formulating such layers as implicit causal self-attention layers. The formulation includes most of their sub-components and is not limited to a specific part of the architecture. The framework compares the underlying mechanisms on similar grounds for different layers and provides a direct means for applying explainability methods. Our experiments show that our attention matrices and attribution method outperform an alternative and a more limited formulation that was recently proposed for Mamba. For the other architectures for which our method is the first to provide such a view, our method is effective and competitive in the relevant metrics compared to the results obtained by state-of-the-art Transformer explainability methods. Our code is publicly available.

https://github.com/Itamarzimm/UnifiedImplicitAttnRepr

## 1 Introduction

The very recent State Space Model (SSM) named Mamba by Gu & Dao (2023) has attracted considerable attention since its recent debut (Lieber et al., 2024; Liu et al., 2024; Zhu et al., 2024; Xu et al., 2024), further establishing it as an efficient and accurate general-purpose model. Like other SSM models (Gu et al., 2021a;b), Mamba is autoregressive during inference and trains efficiently in parallel. Recently, Ali et al. (2024) have highlighted a third aspect of the Mamba model; namely, that it is also an attention model, since it implicitly computes attention.

Attention models can be defined as models that linearly combine the values associated with different elements to create the next set of such associated values. When discussing sequences of tokens, an attention operator considers the values obtained for each token separately, as a hidden representation, and mixes these to obtain a new set of values for each token. The mixing coefficients are also a function of the hidden representations.

Let $X$ be the matrix whose columns are the hidden values associated with each token, and let $\alpha$ be the matrix of mixing coefficients. The set of values of the next layer is initially obtained as $Y = \alpha X$ and it can then undergo other forms of processing, such as nonlinear activations and per-token processing. Given a neural architecture, one can always linearize the mixing operators and write them in the form $Y = \alpha X$ via their first-order approximation. However, to be considered an attention model it is required that $\alpha$ be a function of $X$, which means that the linear operator is data-dependent. This property is shown by Ali et al. (2024) to hold only for the recent selective SSM (S6) , but not for most earlier SSMs. Specifically, for standard state-space layers, it has been demonstrated that they can be linearized into a constant operator, represented by a constant matrix $\alpha$, which is solely controlled by the layer's parameters. However, in the S6 layers, $\alpha$ is influenced by both the input and the layer's parameters.

---

[*]These authors contributed equally to this work.

The implicit attention matrix of Ali et al. (2024) considers the S6 mechanism and ignores the influence of other critical mixer components, such as Conv1D, gate branch, linear layers, and SiLU activations. The formulation we propose in this work incorporates these additional elements and, as we show empirically, leads to improved interpretability results in both computer vision and NLP.

Furthermore, using a similar holistic formulation, we show that S6 is not the only sequence model that implicitly computes attention and that an implicit attention representation can also describe other recent layers, such as RWKV (Peng et al., 2023), Griffin (De et al., 2024), HGRN (Qin et al., 2024b) and more, as illustrated in Figure 1.

To achieve a more accurate representation that better reflects the model's behavior, we employ a composition of multiple components. The concept of composing non-attention layers and representing them as data-controlled linear operators was initially introduced in Hyena (Poli et al., 2023), which attempts to replicate attention capabilities through a composition of two sub-quadratic operators (long convolutions and multiplicative gating). Our formulation differs from this approach in two main ways. First, instead of focusing on replicating attention capabilities, we take the reverse step by demonstrating that, through a sequence of algebraic manipulations, several existing modern gated linear RNNs can be viewed as single implicit attention layers. Second, our goal is to find the most accurate implicit attention representation possible, as it crucial for applications like interpretability. This leads to a significant extension over Hyena's work. For example, while Hyena's matrices are constructed from the two components mentioned above, our implicit attention representation incorporates additional layers, including non-linear operators such as linear layers, activations, short convolutions, and normalization layers. For instance, our formulation for Mamba-2 is built upon six different layers, some of which appear multiple times, resulting in a much more complex outcome. Additionally, while other works explore the relations between non-attention layers and linear attention (Arora et al., 2023), we not aware to any work extending the concept of composition of components similar to us, or apply it to existing modern RNN such as Griffin, Mamba, or link in to interpretability or similar domains.

**Our main contributions** are as follows: (i) We introduce the implicit self-attention representation, unifying Transformers with non-Transformer layers, such as Griffin, RWKV, ReNet, and others. (ii) We refine the approach of Ali et al. (2024) to produce more accurate attention matrices. The previous work focused exclusively on the S6 layer, without considering the gating and Conv1D sub-layers in Mamba, while our representation incorporates all these factors (and additional peripherals in other models) (iii) While "Attention is not Explanation" (Jain & Wallace, 2019), Transformer explainability relies heavily on attention matrices. We demonstrate that our implicit attention representation of non-Transformer models can be used to develop new explainability and interpretability techniques for non-Transformer models, enhancing the community's ability to understand, explore, and manage aspects of robustness, bias, fairness, and safety. As a sample downstream application, we demonstrate excellent out-of-the-box results for attribution-based performance-enhancing techniques. (iv) Finally, our framework facilitates comparisons between Transformers and other recent architectures, by providing a unified attention view and setting the stage for further improvements and insights.

## 2 RELATED WORK

This section describes the scientific context and provides the necessary terminology and symbols for discussing self-attention and selective SSM layers.

**Self-Attention.** Self-attention, a cornerstone of Transformer architectures (Vaswani, 2017), has profoundly influenced recent developments in NLP and computer vision. This mechanism leverages pairwise token interactions to dynamically allocate focus across different parts of the input sequence, assessing the relevance of each token in relation to others. The computational formula is given by:

$$Self - Attention(Q, K, V) = \alpha V, \quad \alpha = \text{softmax}\left(\frac{QK^T}{\sqrt{d_k}}\right) \tag{1}$$

Here, $Q$, $K$, and $V$ denote the queries, keys, and values respectively, with $d_l$ representing the key dimension. Transformers enhance this mechanism by incorporating $H$ parallel attention heads, thus capturing a wider range of dependencies.

**Applications of Attention Matrices.** Attention matrices play a crucial role in Transformers, as multiplying these matrices with value vectors is the core operation that captures interactions between tokens. Beyond this essential role in computing self-attention, they are also used for various

purposes: (i) **Explainability and Interpretability:** Although attention itself is not inherently explainable (Jain & Wallace, 2019), many methods in these domains rely on attention matrices to understand and analyze model behavior (Abnar & Zuidema, 2020; Chefer et al., 2021b;a; Ali et al., 2024) . (ii) **Multi-modal Learning:** Numerous multi-modal learning schemes are based on variations of cross-attention, enabling dependencies to be learned between any pair of tokens of different modalities (Lu et al., 2019; Tan & Bansal, 2019). (iii) **Weakly Supervised Tasks:** Attention matrices can provide a valuable source of supervision, highlighting relevant regions or relationships within the data to guide model learning. These techniques are popular in semantic segmentation (Ru et al., 2022; Wang et al., 2020; Ru et al., 2023), and robustness enhancement (Chefer et al., 2022). Finally, (iv) **Inductive Bias and Regularization Methods:** Since attention matrices represent interactions between tokens, they inherently carry semantic meaning. Therefore, they can be manipulated to incorporate domain knowledge or regulate the model effectively (Li et al., 2018; Attanasio et al., 2022; Bonaldi et al., 2023; Zimerman & Wolf, 2024a).

**S6 Layers and Mamba.** The recently presented selective SSM (Gu & Dao, 2023) (S6) outperforms the previous SSMs and various other architectures in NLP (Anthony et al., 2024; Wang et al., 2024b), vision (Liu et al., 2024; Zhu et al., 2024), graph classification (Wang et al., 2024a; Behrouz & Hashemi, 2024), and more. S6 incorporates a dynamic input-dependent form of the discrete matrices $\bar{A}, \bar{B}$, and $C$, such that for every time-step the SSM employs a different recurrent rule. This technique differs from the previous state-space layers, which use the same set of matrices and recurrent rules for each time step.

Denoting the input sequence by $\hat{x} := (\hat{x}_1, \cdots, \hat{x}_L) \in \mathbb{R}^{L \times D}$ where $\hat{x}_i \in \mathbb{R}^D$, the discrete matrices for time step $i$, namely $\bar{A}_i, \bar{B}_i$, and $C_i$ are defined as:

$$B_i = S_B(\hat{x}_i), \quad C_i = S_C(\hat{x}_i), \quad \Delta_i = \text{Softplus}(S_\Delta(\hat{x}_i)), \quad \bar{A}_i = \exp(\Delta_i A), \quad \bar{B}_i = \Delta_i B_i \quad (2)$$

where $S_B, S_C, S_\Delta$ are linear projection layers, and Softplus is asmooth approximation of ReLU.

The usage of input-dependent time-variant layers adds to the expressivity of the layer (Cohen-Karlik et al., 2025), allowing it to adapt to the input, and potentially captures more complex dependencies. While other input-dependent time-variant mechanisms have been proposed in previous works through gated RNNs, the S5 layer (Smith et al., 2022), or adaptive filtering via input-dependent IIR filters (Lutati et al., 2023), S6 also presents an efficient IO-aware implementation, which is parallelized on GPUs via work-efficient parallel scanners (Blelloch, 1990; Martin & Cundy, 2017).

The Mamba block combines the S6 layer, Conv1D and other elementwise operators. It borrows elements from Gated MLP, and given an input $x := (x_1, \cdots x_L)$, it is computed by:

$$\hat{x} = \text{SiLU}(\text{Conv1D}(\text{Linear}(x))), \quad \hat{z} = \text{SiLU}(\text{Linear}(x)), \quad \hat{y}' = \text{Linear}(\text{Selective SSM}(\hat{x}) \otimes \hat{z}) \quad (3)$$

where $\otimes$ denotes elementwise multiplication.

The entire Mamba model contains $\Lambda$ stacked Mamba blocks with $D$ channels per block. Below, the tensors of the j-th channel in the i-th block are denoted by superscript indices of the form $i, j$.

The vision Mamba architectures (Liu et al., 2024; Zhu et al., 2024) (ViM) follow the vision Transformer (ViT) (Dosovitskiy et al., 2020) but replace the Transformer's self-attention mechanism by two bidirectional Mamba layers, These vision models outperform the standard ViT in terms of accuracy and efficiency, for models of similar parameter counts.

**Gated-Linear RNNs.** RNNs, along with their advanced versions, such as GRU (Chung et al., 2014) and LSTM (Hochreiter & Schmidhuber, 1997), play a fundamental role in deep sequence modeling. Their auto-regressive design decouples sequence length from computational complexity per step, making them highly efficient at decoding. However, they don't scale as effectively as Transformers and often face challenges, such as slow training and vanishing gradients. Recently, linear RNNs have shown improved abilities in capturing long-range dependencies (Gu et al., 2021a; Orvieto et al., 2023) and enhanced scalability (Peng et al., 2024; De et al., 2024). Furthermore, gated linear RNNs deliver surprisingly strong language modeling performance (Mehta et al., 2022; Wang et al., 2022; Peng et al., 2023; Qin et al., 2024b). The most advanced gated linear RNNs include the following variants: (i) RWKV-6 (Peng et al., 2023), which draws inspiration from attention-free Transformers (AFT) (Zhai et al., 2021), (ii) Mamba (Gu & Dao, 2023), which employs selective SSM, (iii) HGRN2 (Qin et al., 2024a), which utilizes state expansion, and (iv) Hawk (De et al., 2024), which is built upon an enhanced variant of the LRU (Orvieto et al., 2023). Other notable

examples include GLA (Yang et al., 2023), GateLoop (Katsch, 2023), and RenNet (Sun et al., 2023). These layers achieve results comparable to Transformers on larger scales, matching well-known models, such as Pythia (Biderman et al., 2023) and LLaMA 2 (Touvron et al., 2023). Moreover, several studies show that hybrid models combining attention mechanisms with gated linear RNNs can be complementary (De et al., 2024; Lieber et al., 2024; Poli et al., 2024; Ma et al., 2022; Baron et al., 2023; Fu et al., 2022), enhancing both approaches. Despite these successes, interpretability and explainability techniques for these models remain relatively unexplored.

## 3 METHOD

In this section, we present a general and holistic data-control linear operator representation that can be applied to (at least) many of the recent non-Transformer architectures and which incorporates all components of the architecture. Our objective is to describe each layer in the form of $y = \tilde{\alpha}x$ such that x and y is the input and output respectively, and $\tilde{\alpha} = f(x; \Theta_{\text{arch}})$ is an attention matrix controlled by the parameters of the model and the input. Sec. 3.1 formulates the entire Mamba and Mamba-2 (Dao & Gu, 2024) architectures as a data-control linear operator, incorporating subcomponents such as Conv1D, gate branches, normalizations and activations. Subsequently, Sections. 3.2 and 3.3 extend our approach to other architectures, such as Griffin (De et al., 2024) and RWKV (Peng et al., 2023). Additionally, in Appendix A, we present how to extract holistic data-controlled attention matrices for RetNet (Sun et al., 2023) and HGRN (Qin et al., 2024b).

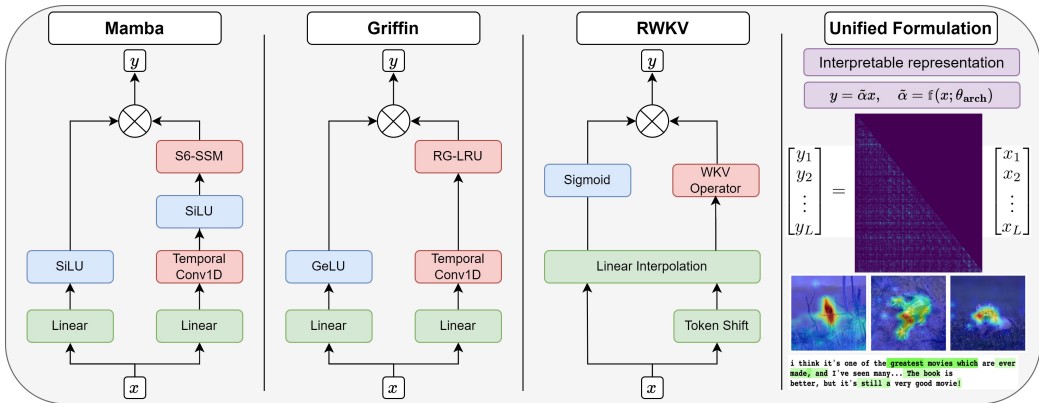

Figure 1: Unified and Interpretable Formulation of Attention-Free Architectures via Attention Matrices: **(Left)** Schematic overview of the architectures of Mamba, Griffin, and RWKV. **(Right)** A new view of those layers that rely on implicit attention. Our perspective enables the generation of implicit attention maps, offering valuable applications in areas such as Explainable AI.

### 3.1 FORMULATION OF MAMBA VIA ATTENTION MATRICES

Mamba can be formulated in a way that separates the components that mix channels from those that mix tokens:

$$\text{Mamba}(x) = \text{Linear}_3\Big( \text{SILU}(\text{Linear}_2(\text{Linear}_1(x))) \otimes \text{S6}(\text{SILU}(\text{Conv1D}(\text{Linear}_1(x)))) \Big) \quad (4)$$

Since $\text{Linear}_1$ and $\text{Linear}_3$ do not mix tokens, they are less relevant to our representation (similar to the MLP layers in Transformers), and we consider the following simplified expression:

$$\text{Mamba}(x) = \Big( \text{SILU}(\text{Linear}_2(x)) \Big) \otimes \Big( \text{S6}(\text{SILU}(\text{Conv1D}(x))) \Big) \quad (5)$$

Replacing the element-wise gating multiplication with matrix multiplication leads to:

$$\text{Mamba}(x) = \text{diag}\Big( \text{SILU}(\text{Linear}_2(x)) \Big)\Big( \text{S6}(\text{SILU}(\text{Conv1D}(x))) \Big) \quad (6)$$

The S6 layer can be formalized as a data-control linear operator (see Eq. 12 in (Ali et al., 2024)):

$$\text{S6}(x) = \hat{\alpha}x, \quad \hat{\alpha}_{i,j} = C_i\Big(\Pi_{k=j+1}^i \bar{A}_k\Big)\bar{B}_j \quad (7)$$

By plugging Eq. 7 into Eq. 6 and since $\text{SILU}(x) = \text{Sigmoid}(x) \cdot x$:

$$\text{Mamba}(x) = \underbrace{\text{diag}\Big( \text{SILU}(\text{Linear}_2(x))\Big)}_{W'_x \in \mathbb{R}^{L \times L}, \quad (\text{gate})} \hat{\alpha} \underbrace{\text{diag}\Big(\text{Sigmoid}(\text{Conv1D}(x))\Big)(\text{Conv1D}(x))}_{Z_{x'} \in \mathbb{R}^{L \times L}, \quad (\text{Conv \& Act})} \quad (8)$$

Recall that causal Conv1D layer with filter $f = (f_1, \cdots, f_{\hat{L}})$ can be converted into a matrix form by arranging shifted copies of the filter into rows, forming a convolution matrix $M$. This matrix is then multiplied by the input sequence to produce an output, where each element represents the dot product of the filter and a corresponding segment of the input.

By plugging the convolution matrix $M$ and the gate matrix $W'_x$ into Eq. 8, we get:

$$\text{Mamba}(x) = W'_x \hat{\alpha} Z_{x'} M x = Hx, \quad H = W'_x \hat{\alpha} Z_{x'} M \quad (9)$$

Therefore, the entire Mamba layer can be viewed as a data-control linear operator, which implicitly parameterizes the per-channel implicit attention matrices through the parameters of the S6 layer, the Conv1D filter, the linear layer in the gate branch, and is controlled by the input $x$

**Mamba-2** This architecture builds upon Mamba by introducing two key enhancements relevant to our formulation: (i) incorporating the concept of multiple heads via a multi-input SSM, and (ii) applying additional normalization (GroupRMSNorm) after the multiplicative gating.

The first modification can be handled by broadcasting parts of the equations across different attention heads. For the second modification, we first compute the per-head statistics necessary for Group Normalization and pack them into a diagonal matrix.

$$\mu_h = \frac{1}{d} \sum_{i=1}^{d} x_h[i], \quad \sigma_h = \epsilon + \sqrt{\frac{1}{d} \sum_{i=1}^{d} (x_h[i] - \mu_h)^2}, \quad \mathbb{N} = \text{diag}\left( \frac{1}{\sigma_1}, \cdots, \frac{1}{\sigma_h}, \cdots, \frac{1}{\sigma_H} \right) \quad (10)$$

where $x_h[i]$ denotes the $i$-th feature of head $h \in [H]$, $d$ is the dimensionality of each head, and $\epsilon$ is a small constant added for numerical stability in GroupRMSNorm.

The matrix $\mathbb{N}$ allows us to represent the GroupRMSNorm operator via matrix multiplication such that $\mathbb{N}x = \text{GroupRMSNorm}(x)$ (where $\mathbb{N}$ is augmented across groups). Thus, we plug these modifications into our formulation of Mamba (Equation 9), obtaining the following implicit-attention formulation for Mamba-2:

$$\text{Mamba-2}(x) = \mathbb{N}W'_x \hat{\alpha} Z_{x'} M x = Hx \quad (11)$$

## 3.2 FORMULATION OF GRIFFIN VIA ATTENTION MATRICES

The component that captures interactions between tokens in Hawk and Griffin (regardless of self-attention) is the temporal mixing block, which is built on top of a Real-Gated Linear Recurrent Unit (RG-LRU), Conv1D, and gating. It can be formalized as follows:

$$y = \text{Linear}_3\Big( \Big(\text{GELU}(\text{Linear}_1(x'))\Big) \otimes \Big(\text{RG-LRU}(\text{Conv1D}(\text{Linear}_2(x')))\Big)\Big) \quad (12)$$

We first rearrange the linear layers and replace elementwise gating with matrix multiplication:

$$x = \text{Linear}_2(x'), \quad y = \text{Linear}_3\Big( \text{diag}\Big(\text{GELU}(\text{Linear}'_1(x))\Big) \Big(\text{RG-LRU}(\text{Conv1D}(x))\Big)\Big) \quad (13)$$

Note that $\text{Linear}'_1 := \text{Linear}_1 \text{Linear}_2$ and $\text{Linear}_3$ do not mix tokens and can therefore be omitted. By substituting Conv1D with matrix multiplication using a causal convolution matrix $M$, we derive:

$$y = \text{diag}\Big(\text{GELU}(\text{Linear}'_1(x))\Big) \Big(\text{RG-LRU}(Mx)\Big) \quad (14)$$

RG-LRU is defined by the following recurrent rule:

$$r_t = \sigma(W_a x_t + b_a), \quad i_t = \sigma(W_x x_t + b_x), \quad a_t = a^{cr_t}, \quad h_t = a_t \otimes h_{t-1} + \sqrt{1 - a_t^2} \otimes (i_t \otimes x_t) \tag{15}$$

This linear recurrent rule can be converted to a matrix form as follows:

$$h = \tilde{\alpha}x, \quad \begin{bmatrix} h_1 \\ h_2 \\ \vdots \\ h_L \end{bmatrix} = \begin{bmatrix} \sqrt{1 - a_1^2} \otimes i_1 & 0 & \cdots & 0 \\ a_2\sqrt{1 - a_1^2} \otimes i_1 & \sqrt{1 - a_2^2} \otimes i_2 & \cdots & 0 \\ \vdots & \vdots & \ddots & 0 \\ \Pi_{k=2}^L a_k \sqrt{1 - a_1^2} \otimes i_1 & \Pi_{k=3}^L a_k \sqrt{1 - a_2^2} \otimes i_2 & \cdots & \sqrt{1 - a_L^2} \otimes i_L \end{bmatrix} \begin{bmatrix} x_1 \\ x_2 \\ \vdots \\ x_L \end{bmatrix} \tag{16}$$

By plugging Eq.16 into Eq.14, we see that the entire temporal mixing block can be formalized as a data-control linear operator:

$$y = \text{diag}\Big(\text{GELU}(\text{Linear}_1'(x))\Big)\tilde{\alpha}Mx = Hx, \quad H = \text{diag}\Big(\text{GELU}(\text{Linear}_1'(x))\Big)\tilde{\alpha}M \tag{17}$$

### 3.3 FORMULATION OF RWKV VIA ATTENTION MATRICES

The time-mixing block of RWKV includes three components: the WKV operator , a gate branch, and a token shift. For simplicity, we will ignore the token shift operation over the values. The simplified RWKV, which maps the input $x_t$ to the output $o_t$ , can be formulated as follows:

$$r_t = W_r \cdot (u_r \otimes x_t + (1 - u_r) \otimes x_{t-1}), \quad k_t = W_k \cdot (u_k \otimes x_t + (1 - u_k) \otimes x_{t-1}), \quad v_t = x_t \tag{18}$$

$$wkv_t = \frac{\sum_{i=1}^{t-1} e^{-(t-1-i)w+k_i} \otimes v_i + e^{u+k_t} \otimes v_t}{\sum_{i=1}^{t-1} e^{-(t-1-i)w+k_i} + e^{u+k_t}}, \quad o_t = W_o \sigma(r_t) \otimes wkv_t \tag{19}$$

where $W_r, W_k, W_o$ are linear projections, and $u, w, u_r, u_k$ are learnable parameters.

Now, we will reformulate the $wkv_t$ operator into a form of causal self-attention:

$$\hat{\alpha}_{i,j} = \begin{cases} \frac{e^{u+k_i}}{\sum_{m=1}^{i-1} e^{-(t-1-m)w+k_i} + e^{u+k_t}} & \text{if } j = i \text{ holds,} \\ \frac{e^{-(i-1-j)w+k_t}}{\sum_{m=1}^{i-1} e^{-(t-1-m)w+k_i} + e^{u+k_t}} & \text{if } j < i \text{ holds,} \\ 0 & \text{otherwise.} \end{cases} \qquad \hat{\alpha}x = wkv \tag{20}$$

Note that $W_o$ does not mix tokens and can therefore be omitted. By plugging Eq. 20 into Eq. 18, and replacing element-wise gating with matrix multiplication, we obtain:

$$o = \text{diag}(\sigma(r))\hat{\alpha}x \tag{21}$$

### 3.4 SHARED PROPERTIES

The proposed formulation for Griffin, Mamba, and RWKV is based on the similarities in the **structure of the architecture**. Our formulation focuses on three main components: (i) the core of the linear attention mechanism (S6 for Mamba, RG-LRU for Griffin, or the WKV operator for RWKV), (ii) a short filter operation implemented via Conv1D in Griffin and Mamba and token shift in RWKV, and (iii) the gate branch, as illustrated in Fig. 1. Additionally, our formulation builds on the following key components: (1) rearranging linear layers and omitting operators that do not influence the mixer components, (2) representing the gate branch, activations, and normalization layers as a data control linear operator via diagonal matrices, (3) unrolling the linear recurrent layer to obtain a token-to-token map, and (4) fusing several cascaded linear operators and ignore biases.

In Appendix E, we elaborate on several insights derived from our formulation of gated-linear RNNs, including the inner dynamics of their parameterization, their expressivity, and the categorization of attention models.

## 4 EXPERIMENTS

To assess the effectiveness of our implicit attention formulation, we perform a comprehensive set of experiments. In Sec. 4.1, we begin by visualizing the implicit attention matrices and the corresponding explainability maps built upon them. In Sec 4.2, we demonstrate that integrating our

improved attention matrices into existing attribution methods results in SoTA interperablity techniques. We further conduct ablations to analyze the contribution of each architectural component to the overall representation. Finally, in Sec. 4.3, we show how our formulation enables the transfer of performance-enhancing techniques, originally designed for other architectures, to gated RNNs.

## 4.1 VISUALIZATIONS

In Fig. 2, we present a comparative visualization of the attention matrices from Mamba, RWKV, Griffin, and Transformers. To enhance clarity, we applied the Softmax function to each row of the attention matrices from the Transformers and conducted min-max normalization on the absolute values of the matrices from the non-Transformer models. In every instance, we used a uniform prompt of size 32. For each model, we examined the attention matrices derived from the standard pre-trained models available in the Hugging Face, including the Recurrent Gemma-2B, RWKV-430M trained on the Pile, and a Mamba-based LLM with 2.8B parameters also trained on the Pile.

As illustrated, the implicit attention matrices of Mamba, Griffin, and RWKV exhibit similarities to those derived from traditional Transformers. Echoing findings from (Ali et al., 2024), we note that dependencies between distant tokens become more apparent in the deeper layers, as shown in the lower rows. Additionally, the matrices from RWKV are characterized by distinct horizontal tiles, whereas those from Mamba display a more continuous structure.

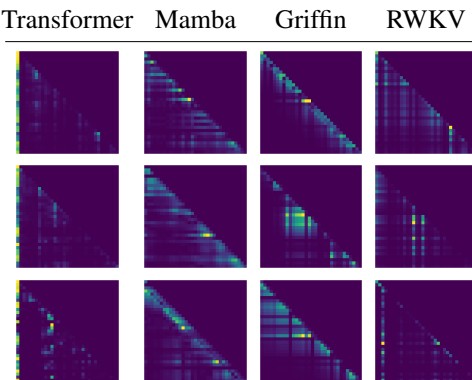

Figure 2: **Hidden Attention Matrices:** Attention matrices of LLMs. Each row represents a different layer within the LLMs, showcasing the evolution of the attention matrices at 25% (top), 50%, and 75% (bottom) of the layer depth.

**Visualization of Explainability Maps.** Sample explainability maps built on top of our implicit attention formulation are resented in Figure 3. This visualization focuses on the rows of the attention maps that are associated with the [CLS] token, as is traditionally done for interpretability purposes. We explore the attention matrices with three explanation methods: raw attention, attention rollout (Abnar & Zuidema, 2020), and attribution following Ali et al. (2024), along with a comparison to the ViT counterparts. Evidently, the explanation methods that are based on our attention formulation (columns e, f, and g) depict much more accurate and sharp maps compared to those of (Ali et al., 2024) and the ViT counterparts. In Fig. 4 we show similar visualizations in the NLP domain. More qualitative results for the NLP domain can be found in Appendix. D.

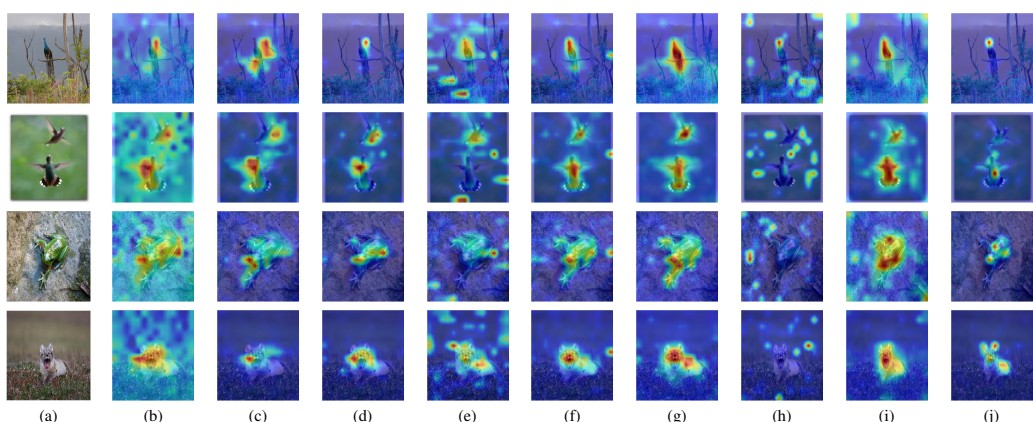

Figure 3: Qualitative results for the different explanation methods for the ViT and ViM, both of small size. (a) The original image, (b) Raw-Attention over ViM, (c) Attention-Rollout over ViM, (d) Mamba-Attribution over ViM, (e) Raw-Attention with our proposed attention over ViM, (f) Attention-Rollout with our proposed attention over ViM, (g) Mamba-Attribution with our proposed attention over ViM, (h) Raw-Attention of ViT, (i) Attention-Rollout for ViT, (j) Transformer-Attribution for ViT. Results for columns (b), (c), and (d) are based on the method of (Ali et al., 2024), and the ViT results on (i), (j) and (k) rely on Chefer et al. (2021b).

Figure 4: Qualitative results for NLP, samples are taken from IMDB movie sentiment classification. In (a), we show the results for the previously proposed Mamba's attention (Ali et al., 2024), (b) our proposed Mamba's attention, and in (c) we show our proposed method over RWKV. In the upper row, we show a negative sentiment, and in the lower row, we show a positive sentiment.

## 4.2 IMPLICIT ATTENTION-BASED ATTRIBUTION

Although empirical evaluation of attribution methods is challenging, this section demonstrates that off-the-shelf techniques, when built on top of our implicit attention formulation, produce SoTA explainability tools. We provide empirical analysis via perturbation and segmentation tests.

**Perturbation Tests.** To assess the faithfulness of explanations, we adopted an input perturbation scheme similar to (Chefer et al., 2021b;a). This method involves systematically masking image pixels based on their predicted relevance from the explanation method. We conducted experiments with both positive and negative perturbations on both NLP and Vision domains. For positive perturbation, a good explanation prioritizes relevant pixels. We expect the model's accuracy (specifically, top-1 accuracy) to gradually decrease as we mask pixels in descending order of relevance (most relevant first). As for negative Perturbation, a robust explanation should maintain model accuracy even when irrelevant pixels are masked. Here, we mask pixels in ascending order of relevance (least relevant first). In both scenarios, we evaluate the explanation quality using the Area-Under-Curve (AUC) metric. AUC considers the model's accuracy as a function of the percentage of masked pixels (ranging from $10\%$ to $90\%$).

The perturbations results for vision models are summarized in Table 1 for various explanation methods under both positive and negative perturbation scenarios on the ImageNet validation set. In the positive perturbation scenario, where lower AUC values indicate better performance, our proposed Mamba's attention method consistently outperforms the other methods. Specifically, our method achieves the lowest AUC values across all explanation methods, with an AUC of $13.264$ for Raw-Attention, $12.830$ for Attn-Rollout, and a notably low $11.350$ for Attribution. In the negative perturbation scenario, where higher AUC values are better, our method shows the best performance, with AUC values of $47.705$ for Raw-Attention, $50.035$ for Attn-Rollout, and $51.310$ for Attribution, outperforming both the method of (Ali et al., 2024) and the counterpart XAI methods for ViT.

Table 1: **Perturbation Tests for Vision.** We present the AUC results (percentages) for the predicted class on the ImageNet validation set. For positive perturbation lower is better, and for negative perturbation higher is better. Previous results by (Ali et al., 2024) denoted by $^\diamond$.

| | Positive Perturbation ↓ | | | Negative Perturbation ↑ | | |
|---|---|---|---|---|---|---|
| | Mamba $^\diamond$ | Mamba Ours | Transformer | Mamba $^\diamond$ | Mamba Ours | Transformer |
| Raw-Attention | 17.268 | 13.264 | 20.687 | 34.025 | 47.705 | 40.766 |
| Attn-Rollout | 18.806 | 12.830 | 20.594 | 41.864 | 50.035 | 43.525 |
| Attribution | 16.619 | **11.350** | 15.351 | 39.632 | **51.310** | 48.089 |

In the NLP domain, we conducted perturbation tests in both the zero-shot and fine-tuned settings. In the zero-shot setting, we utilized pre-trained Mamba-based LLMs with sizes of 1.3B and 2.8B on the ARC-E dataset (Clark et al., 2018), which evaluates the reasoning abilities of LLMs. Results are presented in Table 2 and contain activation and pruning perturbations, as described by Ali et al. (2022). It is shown that our explainability method improves upon the baseline of Ali et al. (2024) for both model sizes. Specifically, in the activation scenario, our method improves results by at least 2.2%, and by over 10% in the pruning settings. Similar trends are also evident in the fine-tuned scenario, see Appendix C for these results for both Mamba and RWKV. Taken together, these results

Table 2: **Perturbation Tests for NLP.** For activation perturbation lower is better, and for pruning perturbation higher is better. Previous results by (Ali et al., 2024) denoted by $^\diamond$.

|  | Activation Perturbation (AUAC) ↑ | | Pruning Perturbation (AU-MSE) ↓ | |
|---|---|---|---|---|
|  | Mamba 1.3B | Mamba 2.8B | Mamba 1.3B | Mamba 2.8B |
| Attribution of (Ali et al., 2024) | 0.915 | 0.918 | 1.765 | 1.239 |
| Our Attribution | **0.934** | **0.939** | **1.588** | **1.082** |

demonstrate that **our attention formulation is much more precise and better reflects the model's behavior** compared to the formulation proposed by Ali et al. (2024). The same phenomenon occurs with the RWKV model, consistently showing that our formulation can lead to SoTA attribution methods for an entire family of models.

**Segmentation Tests.** We evaluated our proposed Mamba's implicit attention mechanism by comparing its generated foreground segmentation maps against ground truth from the ImageNet-Segmentation dataset (Guillaumin et al., 2014). We employed established metrics (pixel accuracy, mean Intersection-over-Union (mIoU), and mean Average Precision (mAP)) aligning with prior works (Chefer et al., 2021b; Nam et al., 2020; Gur et al., 2021). We compared our method against techniques specifically designed for ViM, including the attention-based approach of Ali et al. (2024) and the recent LRP-based method introduced by Jafari et al. (2024). Additionally, as a reference point, we present the results of standard methods applied to ViT.

Results presented in Table 3 demonstrate that our proposed Mamba's implicit attention outperforms both the ViT and the previous method designed for Mamba across all metrics for the three different XAI methods. This superior performance suggests the potential of these maps for downstreaming tasks such as weakly supervised semantic segmentation, and mitigating background bias in classifiers (Chefer et al., 2022).

Table 3: **Segmentation results** on the ImageNet-Segmentation dataset (percent). Higher is better.

| Model | Method | Pixel accuracy ↑ | mAP ↑ | mIoU ↑ |
|---|---|---|---|---|
| DeiT S | Raw-Attention | 59.69 | 77.25 | 36.94 |
| ViM-S | Raw-Attention (Ali et al., 2024) | 67.64 | 74.88 | 45.09 |
| ViM-S | Raw-Attention Ours | **67.66** | **80.00** | **47.28** |
| DeiT-S | Attn-Rollout (Abnar & Zuidema, 2020) | 66.84 | 80.34 | 47.85 |
| ViM-S | Attn-Rollout (Ali et al., 2024) | 71.01 | 80.78 | 51.51 |
| ViM-S | Attn-Rollout Ours | **76.40** | **83.90** | **58.48** |
| DeiT-S | Transformer-Attr (Chefer et al., 2021b) | 79.26 | 84.85 | 60.63 |
| ViM-S | Mamba-Attr (LRP) (Jafari et al., 2024) | 71.19 | 77.04 | 49.98 |
| ViM-S | Mamba-Attr (Ali et al., 2024) | 74.72 | 81.70 | 54.24 |
| ViM-S | Mamba-Attr Ours | **79.60** | **86.40** | **62.51** |

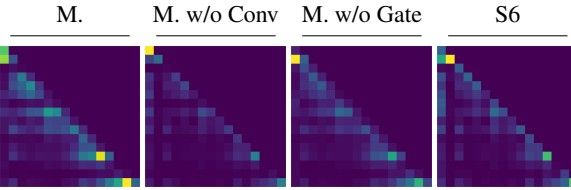

| M. | M. w/o Conv | M. w/o Gate | S6 |
|---|---|---|---|

Figure 5: Comparative visualization of ablated hidden matrices. 'M' for Mamba.

Table 4: **Ablation.** ViM-small for ImageNet Segmentation. Higher is better.

| Method | Pixel Acc. | mAP | mIoU |
|---|---|---|---|
| Full Model | **79.60** | **86.40** | **62.51** |
| w/o SilU | 79.32 | 86.22 | 62.41 |
| w/o Conv1D | 70.01 | 78.87 | 50.64 |
| w/o Gate | 75.11 | 80.12 | 55.78 |
| w/o S6 | 42.99 | 64.13 | 26.01 |
| Only S6 | 72.39 | 80.09 | 53.19 |

**Ablation study.** The architectures we explored implicitly parametrize attention matrices through a composition of several different sub-layers, see Eq.9, 21, and 17. Examples of these sub-layers include linear recurrent layers, gate mechanisms, activations, normalization and other components, such as token-shift or depth-wise convolutions. To better understand the contribution of each of these components, we conduct a sequence of ablation studies. Initially, in Fig. 5, we visualize the

Table 5: Performance of various Mamba-based LLMs on Snarks, CommonsenseQA, and Formal Fallacies datasts. We compare the vanilla model performance, and models employ the Amplified method, with our attribution method or the attribution method of Ali et al. (2024) (denoted by $\diamond$).

| Model Size | Method | Snarks (%) | CommonsenseQA (%) | Formal Fallacies (%) |
|---|---|---|---|---|
| 1.3B | Vanilla | 44.54 | 52.15 | 40.13 |
| 1.3B | AMPLIFY $\diamond$ | 53.11 | 53.55 | 44.28 |
| 1.3B | AMPLIFY (Ours) | **56.15** | **54.72** | **45.22** |
| 2.8B | Vanilla | 48.75 | 53.11 | 44.67 |
| 2.8B | AMPLIFY $\diamond$ | 58.10 | **56.10** | 47.80 |
| 2.8B | AMPLIFY (Ours) | **60.02** | 56.08 | **47.86** |

implicit attention of Mamba, ablating the Conv1D or the gate branch, or focusing solely on the S6 layer. As expected, it seems that the Conv1D causes a smoothing effect, and the Mamba matrices are significantly sharper, with more pixels having non-negligible values compared to those of S6.

In Table 4, we compare several ablation variants of our method. As can be seen, our method, which utilizes all the components of Mamba, achieves a much better score than the ablated versions, illustrating the importance of all components. This experiment reveals that including the S6, Conv1D and gating mechanism is crucial for high performance and reliable representation. However, the activation has a relatively low impact on these aspects. A similar ablation study was conducted for RWKV and presented in Appendix C, demonstrating similar trends.

### 4.3 ATTRIBUTION-BASED PERFORMANCE-ENHANCING TECHNIQUES

To further demonstrate the practical impact of our representation, we show it can enhances model performance. While various attention-based and explainability-based techniques was previously proposed for improving model performance, our focus is on in-context learning (ICL) and weakly supervised semantic segmentation tasks.

To improve ICL, we adopt the AMPLIFY method of Krishna et al. (2024), a prompt engineering technique designed for few-shot ICL, which leverages post-hoc explanation methods. In our experiments, we use the Mamba-790m model as a proxy, following the same evaluation protocol as AMPLIFY, but with an attribution method that relies on attention matrices. We report the vanilla model performance, and the performance with AMPLIFY with our attribution method and the attribution method of Ali et al. (2024). The results are depicted in Tab. 5, show that our method outperforms the baseline across all tested scenarios except one, with an average margin of 1.2 accuracy points over the amplify baseline, and 9.8% over the vanilla baseline.

Detailed results and experimental settings for weakly supervised semantic segmentation are presented in Appendix B.

### 5 CONCLUSIONS

In this study, we have extended the use of self-attention from its traditional role as the core mechanism of Transformers to a representation of neural sequence layers. Our unified framework facilitates the exploration of similarities and differences among non-attention layers, such as Mamba, RWKV, and Griffin, and their interconnections with Transformer architectures. Additionally, it enables the development of innovative explainability techniques for the latest attention-free architectures. Our contributions provide the research community with new tools for analyzing the performance, fairness, and robustness of gated-linear RNN variants, while also identifying their potential vulnerabilities. These advancements set the stage for future improvements and support the implementation of weakly supervised downstream tasks.

Looking ahead, we aim to incorporate additional layers, such as Hyena (Poli et al., 2023), and HGRN2 (Qin et al., 2024a) into our framework, including their vision-specific variants (Duan et al., 2024; Fan et al., 2023; Zimerman & Wolf, 2024b; Spravil et al., 2024). Furthermore, we plan to examine how differences in these architectures are reflected in their self-attention matrices and explore whether such insights can reveal more about the inductive bias inherent in each architecture.

## 6 REPRODUCIBILITY STATEMENT

All of our experiments are conducted using the PyTorch framework on public datasets. Additionally, our code for some of the experiments is included as supplementary, along with a user-friendly interface and notebook demos. Therefore, we consider our empirical results to be reproducible.

## 7 ACKNOWLEDGMENTS

This work was supported by a grant from the Tel Aviv University Center for AI and Data Science (TAD). This research was also supported by the Ministry of Innovation, Science & Technology ,Israel (1001576154) and the Michael J. Fox Foundation (MJFF-022407). The contribution of the first author is part of a PhD thesis research conducted at Tel Aviv University.

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

## A   REPRESENTING ADDITIONAL ARCHITECTURES VIA IMPLICIT ATTENTION

In sec. 3 we present the formulation of Griffin, RWKV, and Mamba via attention matrices. In this section, we extend our method to other layers, such as RetNet (Sun et al., 2023) and HGRN (Qin et al., 2024b).

**RetNet.**    The Retention Network is composed of two primary blocks: (i) the Multi-Scale Retention (MSR) layer and the (ii) FFN layer, which operates independently across tokens. The MSR layer, responsible for token mixing, is built on top of the retention sub-layer and is defined as follows:

$$\text{head}_i = \text{Retention}(X, \gamma_i), \quad \gamma_i = 1 - 2^{-5-i}, \quad Y = \text{GroupNorm}_h(\text{Concat}(\text{head}_1, \cdots, \text{head}_h)) \tag{22}$$

Furthermore, the outputs are scaled using a data-control gate branch, parameterized by a matrix $W_G \in \mathbb{R}^{D \times D}$:

$$\text{MSR(X)} = (\text{swish}(XW_G) \otimes Y) \tag{23}$$

To refine this formulation, we can represent the element-wise multiplication as a matrix multiplication using a diagonal matrix $G = \text{diag}(\text{swish}(XW_G))$. Additionally, per-head statistics can be integrated into $G$. Given that the parallel representation of Retention can be depicted via an attention matrix $R$ (see Eq. 5 in (Sun et al., 2023)), the entire MSR block simplifies to:

$$\text{Retention}(x) = GRx \tag{24}$$

**HGRN.**    The Hierarchically Gated RNN (HGRN) is first defined with the following recurrent rule:

$$\mathbf{f}_t = \text{Sigmoid}\left(\mathbf{x}_t \mathbf{W}_f + \mathbf{b}_f\right) \in \mathbb{R}^{1 \times d}, \quad \mathbf{i}_t = \text{Sigmoid}\left(\mathbf{x}_t \mathbf{W}_i + \mathbf{b}_i\right) \in \mathbb{R}^{1 \times d} \tag{25}$$

$$\mathbf{c}_t = \text{SiLU}\left(\mathbf{x}_t \mathbf{W}_t + \mathbf{b}_z\right) \in \mathbb{R}^{1 \times d}, \quad \mathbf{h}_t = \mathbf{f}_t \otimes \mathbf{h}_{t-1} + \mathbf{i}_t \otimes \mathbf{c}_t \in \mathbb{R}^{1 \times d} \tag{26}$$

where the output of the recurrent $h_t$ is multiplied by $g_t = \text{SiLU}(\text{Linear}(x))$ to produce the output:

$$o_t = g_t \otimes h_t \tag{27}$$

Note that the recurrent rule of the HGRN layer can be computed via an implicit attention represented by a matrix $\alpha_r$ (see Eq. 5 in (Qin et al., 2024b)), as follows:

$$H := (h_1, \cdots, h_L), \quad C = (c_1, \cdots, \quad c_L), \quad H = \alpha_r c \tag{28}$$

Hence, by define $G = \text{diag}(\text{SiLU}(\text{Linar}(x)))$, $G_{act} = \text{diag}(\text{sigmoid}(x))$.

Furthermore, we can rearrange the linear layer such that $W_t$ and $b_z$ will be omitted, and obtain:

$$G_{\text{ACT}} = \text{diag}(\text{Sigmoid}(x)), \quad o = G\alpha_r G_{\text{ACT}}x \tag{29}$$

which is a linear operator characterized by an input-dependent matrix, defined as $G = \text{diag}(\text{Sigmoid}(x))$. The output $o$ is given by $o = G\alpha_r$.

$$\mathbf{f}_t = \text{Sigmoid}\left(\mathbf{x}_t \mathbf{W}_f + \mathbf{b}_f\right) \in \mathbb{R}^{1 \times d}, \quad \mathbf{i}_t = \text{Sigmoid}\left(\mathbf{x}_t \mathbf{W}_i + \mathbf{b}_i\right) \in \mathbb{R}^{1 \times d} \tag{30}$$

$$\mathbf{c}_t = \text{SiLU}\left(\mathbf{x}_t \mathbf{W}_t + \mathbf{b}_z\right) \in \mathbb{R}^{1 \times d}, \quad \mathbf{h}_t = \mathbf{f}_t \otimes \mathbf{h}_{t-1} + \mathbf{i}_t \otimes \mathbf{c}_t \in \mathbb{R}^{1 \times d} \tag{31}$$

where the output of the recurrent $h_t$ is multiplied by $g_t = \text{SiLU}(\text{Linar}(x))$ to produce the output:

$$o_t = g_t \otimes h_t \tag{32}$$

Note that the recurrent rule of the HRGU layer can be computed via an implicit attention represented by a matrix $\alpha_r$ (see Eq. 5 in (Qin et al., 2024b)), as follows:

$$H := (h_1, \cdots, h_L), \quad C = (c_1, \cdots, \quad c_L), \quad H = \alpha_r c \tag{33}$$

Hence, by define $G = \text{diag}(\text{SiLU}(\text{Linar}(x)))$, $G_{act} = \text{diag}(\text{sigmoid}(x))$.

Furthermore, we can rearrange the linear layer such the $W_t, b_z$ will be omitted, and obtain:

$$G_{\text{ACT}} = \text{diag}(\text{Sigmoid}(x)), \quad o = G\alpha_r G_{\text{ACT}} x \tag{34}$$

as requested.

## B  WEAKLY SUPERVISED SEMANTIC SEGMENTATION

In weakly supervised semantic segmentation (WSSS), a common approach involves first training a classifier on image-level labels and then extracting Class Activation Maps (CAMs) for individual images, which highlight regions that the classifier deems relevant to specific classes. The SoTA methods then employ these CAMs as pseudo-masks to train a segmentation decoder.

In this context, we adopt our proposed Mamba-Attr XAI method for vision-Mamba models. We assess its competitiveness against the well-established CAMs in generating pseudo-labels for Transformers. To ensure a fair comparison, we fine-tune both DeiT-Small and ViM-Small models under identical conditions over the Pascal-voc 2012 (Everingham et al., 2010) dataset, excluding multi-scale training, inference, or any other modifications. This controlled setting isolates the influence of our Mamba-Attr method on the quality of the generated pseudo-labels.

The results are presented in Table 6. Evidently, Mamba-Attr XAI method achieves competitive results surpassing the baseline approach of Class Activation Maps (CAMs) without any additional modifications. This is evident in the mean Intersection-over-Union (mIoU) score, where Mamba-Attr (52.11%) outperforms the CAM of DeiT-Small (35.99%) by a sizable gap. While Mamba-Attr does not reach the state-of-the-art performance of Toco (Ru et al., 2023) (61.10%), it achieves, out of the box, a substantial improvement over CAM and comes surprisingly close to this much more elaborate multi-phase learning method which utilizes multiple loss terms specifically designed to enhance the quality of the initial CAM map. These results suggest that Mamba-Attr XAI offers a powerful and efficient solution for WSSS tasks with vision-Mamba models.

Table 6: Evaluation and comparison of the pseudo-labels for the different classes in Pascal-voc 2012 (Everingham et al., 2010). Results are in mIoU.

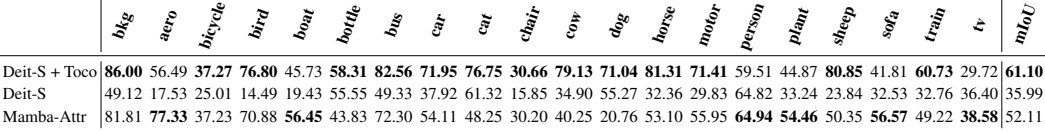

| | bkg | aero | bicycle | bird | boat | bottle | bus | car | cat | chair | cow | dog | horse | motor | person | plant | sheep | sofa | train | tv | mIoU |
|---|---|---|---|---|---|---|---|---|---|---|---|---|---|---|---|---|---|---|---|---|---|
| Deit-S + Toco | 86.00 | 56.49 | 37.27 | 76.80 | 45.73 | 58.31 | 82.56 | 71.95 | 76.75 | 30.66 | 79.13 | 71.04 | 81.31 | 71.41 | 59.51 | 44.87 | 80.85 | 41.81 | 60.73 | 29.72 | 61.10 |
| Deit-S | 49.12 | 17.53 | 25.01 | 14.49 | 19.43 | 55.55 | 49.33 | 37.92 | 61.32 | 15.85 | 34.90 | 55.27 | 32.36 | 29.83 | 64.82 | 33.24 | 23.84 | 32.53 | 32.76 | 36.40 | 35.99 |
| Mamba-Attr | 81.81 | 77.33 | 37.23 | 70.88 | 56.45 | 43.83 | 72.30 | 54.11 | 48.25 | 30.20 | 40.25 | 20.76 | 53.10 | 55.95 | 64.94 | 54.46 | 50.35 | 56.57 | 49.22 | 38.58 | 52.11 |

## C  PERTURBATION EXPERIMENTS FOR NLP

In this section, we present results for the perturbation test in the NLP domain with fine-tuned classifiers. In this setting, we fine-tune the last layers of various LLMs and append the [CLS] token to all samples to generate explanation maps, similar to the methods used in vision models.

In Figure 6, we present perturbation results for both positive and negative settings. We utilize Mamba[1], RWKV[2] and BERT[3] models as the models of interest and fine-tune them on the IMDB sentiment classification dataset, employing the [CLS] token for all samples. Subsequently, we evaluate the explanation quality using a similar procedure as proposed in (Ali et al., 2024) for the perturbation experiment.

The results reveal that Mamba-attr, based on our new attention formulation, achieves superior AUC for both negative and positive perturbations compared to the previous attention formulation by (Ali

---

[1] https://huggingface.co/trinhxuankhai/mamba_text_classification
[2] https://huggingface.co/BlinkDL/rwkv-2-pile-430m
[3] https://github.com/hila-chefer/Transformer-Explainability

et al., 2024). Additionally, our unified attention formulation is effective for RWKV models, yielding comparable results to those of Mamba and BERT.

Moreover, as an ablation, the first column of Figure 6 demonstrates that including the gate branch, as presented in our full method, consistently improves performance.

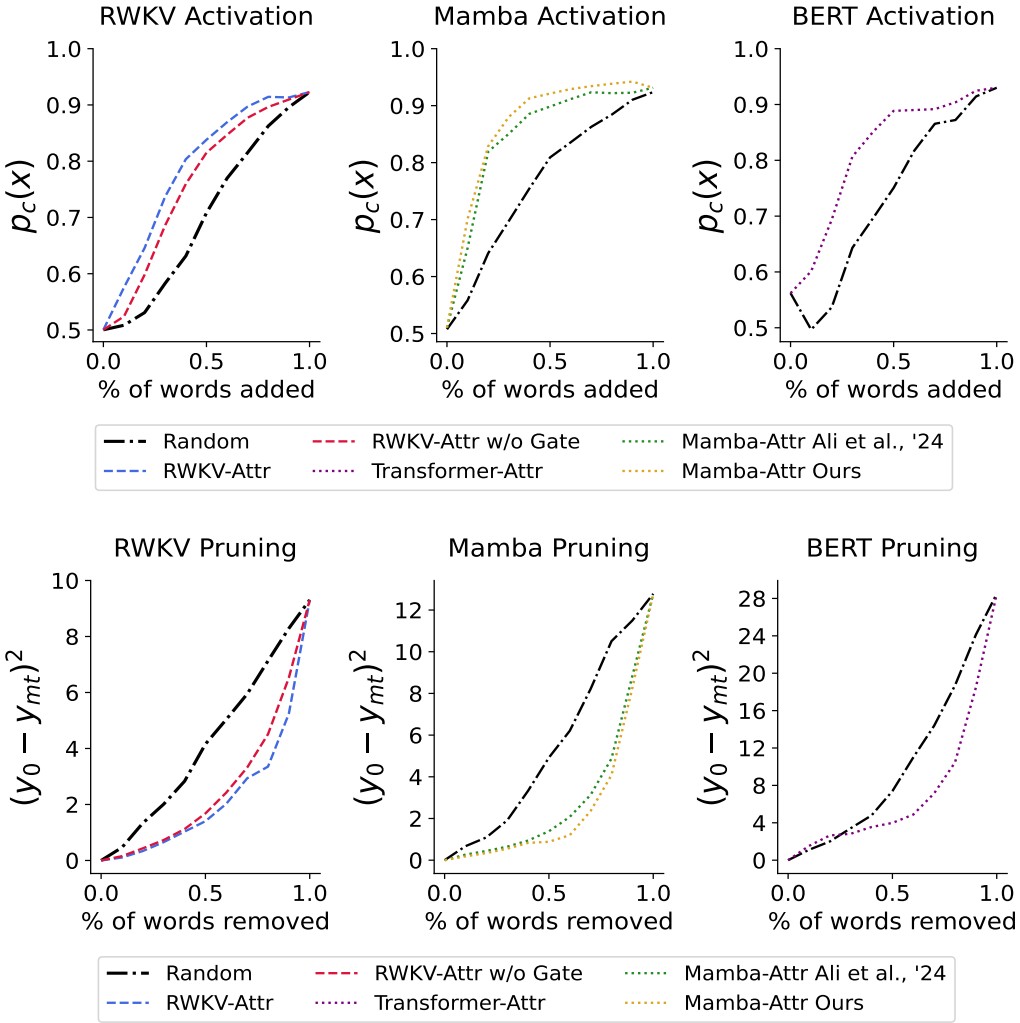

Figure 6: Evaluation of explanations using input perturbations. Results for IMDB activation task (top row) in which the most relevant words are added first, and for IMDB pruning task (lower row) in which the words of least relevance are removed first. Results are shown for 3 different models: RWKV, Mamba, and BERT, respectively.

## D  ADDITIONAL QUALITATIVE RESULTS FOR NLP

Additional NLP results obtained on IMDB dataset are presented in Figure 7. In panel (a), we show the results for the previously proposed Mamba's attention (Ali et al., 2024). Panel (b) shows our proposed Mamba's attention. Lastly, panel(c) presents our proposed method over RWKV. In red, we show a negative sentiment, and in blue, we show a positive sentiment.

As can be seen from these qualitative results, the explanation maps generated by our new attention formulation exhibit sparser and more accurate heatmaps of relevant words than those of Ali et al. (2024), aligning with the desired properties of XAI methods. Similarly, the results for RWKV models show comparable success to those of Mamba.

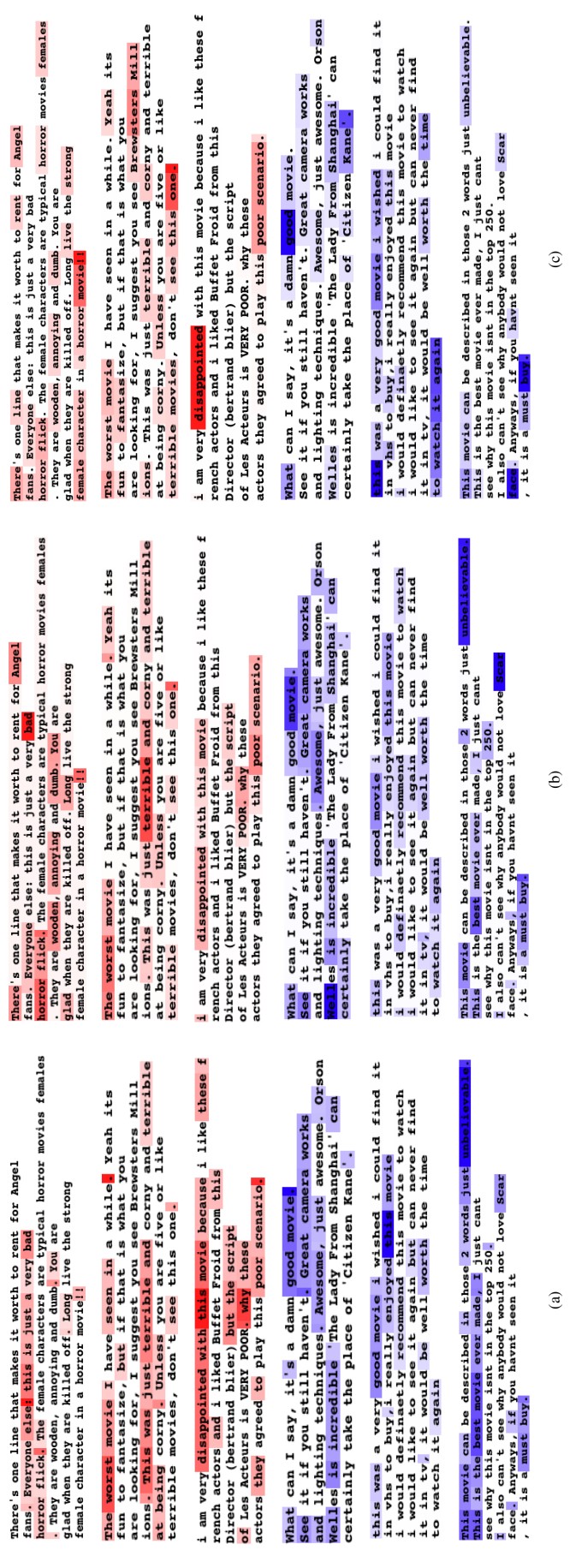

Table 7: Additional qualitative results on the IMDB dataset. (a) The Mamba attention of Ali et al. (2024). (b) Our Mamba attention method. (c) Our RWKV attention.

# E   UNDERSTANDING THE INNER DYNAMICS OF GATED-LINEAR RNNS

Our unified formulation, presented in Equations 9, 11, 17, 21, 24, 34, reveals that several modern gated-linear RNNs can be viewed as data-controlled linear operators. Specifically, these models can be expressed in the form $Y = AX$ where $A$ is an implicit attention matrix dynamically controlled by the input data. This matrix $A$ is parameterized through various sub-layers, including linear recurrent layers, Conv1D, linear layers, gating mechanisms, activations, and normalization layers.

Beyond the applications discussed in Section 2, this formulation sheds light on the inner dynamics and design choices inherent in the parameterization of these layers. It suggests that the entire mixing block in these models, including several sub-layers, implicitly parameterizes **complex and expressive** attention matrices. This implies that these components collectively enable the model to efficiently and implicitly express thousands of data-dependent (where transformers of the same size typically have around a few dozen), complex attention heads.

Given the distinct behaviors of these sub-layers, we propose that they incorporate specific **inductive bias** into the model. In particular, the Conv1D layers promote the inductive bias for attention matrices to be relatively local and smooth. Similarly, the gating mechanisms, linear layers, and activations allow the model to capture sparse and complex (non-linear) features that depend on the input data. Finally, the LayerNorm can be reinterpreted as a per-row data-dependent scaling of the attention matrix, analogous to the role of the softmax function in Transformers.

Our unified view also enables us to **categorize** attention models into three distinct types based on their expressivity and efficiency: (i) **Explicit Attention Models**: Models like the Transformer compute the attention matrix A explicitly during the forward pass, resulting in quadratic space complexity with respect to sequence length. The attention weights are directly calculated from the input data, allowing straightforward interpretability of how each element attends to others. (ii) **Implicit Attention Models:** Models such as RWKV, Mamba, and Griffin fall into this category. Although they do not compute the attention matrix explicitly during the forward path, our formulation demonstrates that their operations can be interpreted as implicit attention mechanisms. This implicit computation allows for greater efficiency with sub-quadratic complexity. To further enhance the expressivity and inductive bias of these implicit attention matrices, several sub-layers can be utilized. (iii) **Attention Tensor Models:** In these models, the operation cannot be represented simply as $Y = AX$ with being a matrix. Instead, $A$ is a higher-order tensor, capturing more complex interactions beyond pairwise relationships. The S5 model serves as an example, where the data-control linear operator does not mix tokens for each channel independently.

This categorization highlights the trade-offs and design considerations in different architectures. In particular, explicit attention models are relatively more expressive but may suffer from higher computational costs. In contrast to implicit attention models that balance between efficiency and expressivity. Lastly, the attention tensor based models provide the capacity to model more complex interactions at the expense of increased computational complexity and less interpretable representation.

This perspective not only enhances our theoretical understanding but also has practical implications. It suggests that critical design choices in these models, such as the inclusion of Conv1D layers, gating mechanisms, and normalization techniques are critical for shaping the implicit attention patterns. Future work can leverage this insight to optimize architectures further, potentially leading to more efficient and expressive sequence models.

