# OpenReview forum: "Explaining Modern Gated-Linear RNNs via a Unified Implicit Attention Formulation"
_ICLR.cc/2025/Conference — ICLR 2025 Poster_

### Official Review · Reviewer_PBPq · 2024-10-28

**Soundness:** 3
**Presentation:** 4
**Contribution:** 3
**Rating:** 8
**Confidence:** 3

**Summary:**

This paper tries to provide an implicit self-attention formulation for most state-of-the art non-transformer sequence models (known as Gated Linear RNNs) such as Mamba, Griffin, RWKV, and ReNet. In this way, it can exploit techniques used in attention explainablity to explain these new models.

Compared to the closest work (Ali et al, 2024), which only formulates the S6 layer in Mamba, the main contribution of paper is:
- Formulating more layers with implicit self-attention and propose a unified and more accurate attention-based representation for all the SOTA gated linear RNNs.

Other contributions include:
- Introducing new explainablity technique for these models leveraging the self-attention formulation
- Showing performance of their explanations and attributions by purturbation testing and segmentation.
- Showing their proposed formulation can give attributions which can further used in some performance-enhancing techniques (based on in-context learning) for large language models.

**Strengths:**

- Covering multiple models including most popular modern non-transformer sequence models.
- Evaluating the performance of the resulting attributions in multiple quantitate experiments and down-stream tasks (across both vision and NLP).
- Showing the impact of various sub-layers in the ablations study

**Weaknesses:**

- In their ablation study, the authors could discuss the trade-off between the time explainability and more accurate formulation/explainability.

- The main baseline paper (Ali et al, 2024) has not been published yet. So, it is hard to evaluate this paper. Actually, the performance of the model in downstream tasks such as segmentation and attribution-based performance-enhancement helped me to have better evaluation of the proposed method.

**Questions:**

mentioned above.

---

> ### Author Response · Authors · 2024-11-16
>
> We are grateful for your detailed review, which has guided us in improving our paper.
>
> > In their ablation study, the authors could discuss the trade-off between the time explainability and more accurate formulation/explainability.
>
> We thank the reviewer for raising this insightful question, which provides an excellent opportunity to further extend our research!
>
> We agree that exploring the trade-off between efficiency and the accuracy of explainability is both important and interesting. Upon analyzing the efficiency results of the variants in Table 4 ("Ablation"), we observe that the differences between the methods are relatively minor. This is because all the variants rely on the S6 layer, which remains the primary bottleneck in terms of time complexity when compared to the Conv1D, gate, and activation components. These findings further highlight the effectiveness of our method compared to the work of Ali et al (2024).
>
> To address this, we are exploring a more efficient formulation that omits the time-consuming S6 layer and instead relies only on Conv1D, gates, and activations. This approach can be further enriched by selectively incorporating a subset of S6 parameters. The motivation stems from the observation that the full S6 layer significantly contributes to computational overhead. Furthermore, this exploration could provide valuable insights into the role of the S6 layer within the Mamba block and its critical contribution to XAI.
>
> > The main baseline paper (Ali et al, 2024) has not been published yet. So, it is hard to evaluate this paper. Actually, the performance of the model in downstream tasks such as segmentation and attribution-based performance-enhancement helped me to have better evaluation of the proposed method.
>
> Thank you for your feedback and for highlighting the importance of downstream tasks in evaluating our proposed method.
>
> In response to your comment, we have included a comparison with Mamba-LRP [1], which was recently accepted to NeurIPS 2024. As shown in Table 3 of the revised manuscript, our method consistently outperforms the LRP approach across all tested benchmarks. Furthermore, our approach surpasses SoTA explainability methods designed for transformers, such as those presented in Chefer et al. (2020), which leverage highly optimized hybrid techniques that combine LRP scores and attention activation maps. This performance demonstrates the effectiveness of our unified framework in delivering high-quality explanations that accurately reflect the model's behavior across a range of modern gated-linear RNN architectures.
>
> ___
> [1] MambaLRP: Explaining Selective State Space Sequence Models. Jafari et al. NeurIPS 2024

---

> > ### Comment · Reviewer_PBPq · 2024-11-23
> > **Response to Rebuttal**
> >
> > Thanks for your response. I keep my rating as is.

---

> > > ### Author Response · Authors · 2024-11-26
> > >
> > > We thank the reviewer again for their feedback.
> > >
> > > > In their ablation study, the authors could discuss the trade-off between the time explainability and more accurate formulation/explainability (Cont.)
> > >
> > >
> > > As promised, we explored this idea over the past week. Specifically, we tested an efficient variant of our method that excludes the S6 layer, the most computationally expensive operator in our formulation and the primary contributor to the FLOP count. Since the S6 layer is integral to the implicit attention formulation, it is expected to be critical for achieving high performance. Indeed, without it, the poor results outweigh the efficiency gains. For example, in ImageNet segmentation with ViM, Pixel Accuracy dropped significantly from 79.60 to 42.99 (see the blue row in Table 4 of the revised manuscript). While this variant is 2.5 times faster than our full method, the substantial performance degradation makes it unviable.
> > > Given that the S6 layer dominates the computational complexity of our method, exploring other efficiency trade-offs or alternative variants is unnecessary, as their runtime differences would be negligible.

---

### Official Review · Reviewer_rDjF · 2024-11-01

**Soundness:** 3
**Presentation:** 3
**Contribution:** 3
**Rating:** 6
**Confidence:** 2

**Summary:**

This paper writes out gated recurrent architectures such as Mamba, RKWV, and Griffin as causal self-attention layers. The main objective of this is to increase interpretability, which is tested through perturbation and segmentation tests.

**Strengths:**

In my view, this is a pretty complete paper. From my understanding, the authors present an extension of the paper Ali et al. (2024) to include additional architectural components in the linearization and not just the S6 mechanism considered before. The authors show that the model improves interpretability through visualizations and a set of both perturbation and segmentation tests. The ablation gives quite a lot of strength to their arguments, but I am not so familiar with these types of explainability results so I am not able to comment on the details.

**Weaknesses:**

While the authors have explored the interpretability side of things extensively, I was wondering if it would be worth comparing the performance of the linearized models compared to its recurrent counterparts when trained on some small datasets?

**Questions:**

See above

---

> ### Author Response · Authors · 2024-11-16
>
> We thank the reviewer for their comments.
>
> > While the authors have explored the interpretability side of things extensively, I was wondering if it would be worth comparing the performance of the linearized models compared to its recurrent counterparts when trained on some small datasets?
>
> Our unified implicit attention formulation serves as a reinterpretation of the original models rather than a structural modification. Consequently, it produces equivalent behavior, apart from minor numerical differences and slower computation.
>
> Beyond enhancing interpretability, in Section 4.3 and Appendix B, we demonstrate that our formulation can be used to improve performance on weakly supervised semantic segmentation tasks and enhance ICL capabilities. Moreover, the revised manuscript also includes a section discussing the theoretical insights arising from our formulation (please see details in the main response).

---

> ### Comment · Reviewer_rDjF · 2024-11-26
> **Reply to authors**
>
> Thank you for your clarifications and the added details. I am keeping my score as is. I think that the paper is a good one, but my low familiarity with the topic of interpretability prevents me from engaging with the authors further. Thank you for taking the time to reply to my comments and I apologize for not providing further feedback.

---

### Official Review · Reviewer_JKNZ · 2024-11-04

**Soundness:** 3
**Presentation:** 3
**Contribution:** 2
**Rating:** 5
**Confidence:** 3

**Summary:**

This paper introduces a unified framework that reformulates various gated recurrent neural network (RNN) architectures, such as Mamba, RWKV, and Griffin, into implicit causal self-attention layers. This reinterpretation aims to make these models more interpretable by constructing attention-like matrices for use in visual and NLP explainability tasks. Experimental evaluations demonstrate that this approach achieves competitive performance in robustness and attribution metrics, though prior work has already suggested that certain gated models, including Mamba, are attention-based.

**Strengths:**

The paper offers a clear explanation of how implicit attention could be used to interpret gated RNNs, making it accessible to readers interested in explainability across model types.

By applying the framework to both NLP and vision tasks, the authors demonstrate its cross-domain relevance.

**Weaknesses:**

Previous work has already conceptualized models like Mamba as attention-like, meaning that simply reinterpreting these gated RNNs under an implicit attention framework may not be largely novel.

The paper does not thoroughly compare its implicit attention framework with existing interpretability tools for gated RNNs.

**Questions:**

A critical question is: given that previous work has noted the attention-like properties of models such as Mamba, what specific benefits does the implicit attention framework offer over these prior interpretations?

What are the current explainability methods or metrics for modern gated-linear RNNs and how's the comparison between them and attention matrices?

---

> ### Author Response · Authors · 2024-11-16
>
> Thank you for your effort and for taking the time to help us improve our paper.
>
> > Previous work has already conceptualized models like Mamba as attention-like, meaning that simply reinterpreting these gated RNNs under an implicit attention framework may not be largely novel.. what specific benefits does the implicit attention framework offer over these prior interpretations? ….  A critical question is: given that previous work has noted the attention-like properties of models such as Mamba, what specific benefits does the implicit attention framework offer over these prior interpretations
>
> We acknowledge that prior work, such as Ali et al. (2024), has demonstrated that gated-linear RNN layers can be interpreted through an implicit attention framework. However, **their formulation overlooks key components**, including Conv1D layers, activations, linear layers, gate branches, and normalizations, as well as a significant portion of the FLOPs and parameters. In contrast, our formulation incorporates these components, **offering a more comprehensive and precise representation** of the model’s behavior. This is achieved through a series of algebraic manipulations.
>
> Our approach addresses this gap by unifying most of Mamba's architecture with **additional architectures** under the attention framework.
>
> We empirically validate that this unification provides a more accurate perspective, leading to superior XAI techniques. For instance, as shown in Table 1, our method surpasses that of Ali et al. by an average of 25%, highlighting a substantial advancement.
>
> > The paper does not thoroughly compare its implicit attention framework with existing interpretability tools for gated RNNs.
>
> Model-specific XAI methods are regarded as the most effective tools for providing meaningful explanations for deep learning architectures. For gated-linear RNNs, general-purpose interpretability tools have not been extensively explored, and to the best of our knowledge, the only prior approach specifically addressing this problem was proposed by Ali et al. (2024). We comprehensively benchmarked our method against theirs in Tables 1–5, demonstrating consistent improvements across all evaluated metrics.
>
> Recently, Jafari et al. [1] introduced an LRP-based explainability method for Mamba. In response to your comment, we conducted additional experiments to compare our approach with theirs. These results have been incorporated into the revised manuscript (see Table 3 in the revised manuscript and the shared response). As shown, our method achieves substantial improvements over Jafari et al.’s approach, setting a new SoTA for explainability in Mamba and further highlighting the versatility and broader applicability of our formulation.
> If the reviewer is aware of any additional relevant work, we would be happy to include it in our analysis to further enrich this study.
>
> > What are the current explainability methods or metrics for modern gated-linear RNNs and how's the comparison between them and attention matrices?
>
> As mentioned in our previous response, the field of explainability for modern gated-linear RNNs is still emerging, with relatively few established methods. To the best of our knowledge, the most notable works in this area are Ali et al. (2024) and the very recent Mamba LRP [1], both of which are limited to the Mamba architecture. Our revised manuscript includes a comprehensive comparison with these approaches (please see Table 3 and the shared response). This comparison, detailed in Section 4 and summarized in Tables 1-5, demonstrates that our method consistently outperforms these techniques across various metrics. These results underscore the effectiveness and generality of our proposed framework, which also extends beyond Mamba to provide explainability for a broader range of gated-linear RNN architectures such as RWKV.
>
> ___
> [1] MambaLRP: Explaining Selective State Space Sequence Models. Jafari et al. NeurIPS 2024

---

> > ### Author Response · Authors · 2024-12-01
> >
> > Dear Reviewer JKNZ,
> >
> > First, thank you for the time and effort you've dedicated to reviewing our submission. We've done our best to address your concerns, including clarifying our contributions over previous work and adding comparisons with other existing interpretability tools for gated RNNs, such as MambaLRP [1] (see head-to-head comparisons in Table 3 of the revised manuscript). As we near the end of the discussion period, we’d love to hear your thoughts on whether these updates address your critiques.

---

### Official Review · Reviewer_xsWF · 2024-11-09

**Soundness:** 3
**Presentation:** 2
**Contribution:** 3
**Rating:** 6
**Confidence:** 2

**Summary:**

The paper studies the problem of sequence modelling. The authors aim to provide a unified framework of the recent attention-free methods such as Mamba and RWKV. The paper presents empirical results to validate the proposed unified framework.

**Strengths:**

- The unified framework makes it easier to study and compare the different sequence modelling algorithms.
- It is important for the community to learn about such work.
- The experimental results are interesting.

**Weaknesses:**

Although it is hard to evaluate such approaches empirically, interpretability-based metrics are not very conclusive in general.

**Questions:**

- What is the best method to evaluate the unified framework beside the interpretability analysis?

---

> ### Author Response · Authors · 2024-11-16
>
> Thank you for taking the time to provide such insightful feedback.
>
> > Although it is hard to evaluate such approaches empirically, interpretability-based metrics are not very conclusive in general.
>
> While we agree that interpretability metrics alone can sometimes be inconclusive, we believe our empirical evidence provides substantial support for our claims, as outlined below:
>
> **(i) Broader Applicability Beyond Interpretability:** In Section 4.3 and Appendix B, we demonstrate that our formulation extends beyond interpretability, proving valuable in improving performance on weakly supervised semantic segmentation tasks and enhancing in-context learning capabilities. These results underscore the broader utility and robustness of our approach across various tasks.
>
> **(ii) Significant and Robust Improvement Without Tuning:** By integrating our attention representation directly with the existing framework proposed by Ali et al. (2024), without any additional modifications to hyperparameters or other parts of the method, we observe a substantial improvement. This result highlights the strength of our method, as it consistently outperforms Ali et al. by a large margin. Furthermore, while the method of Ali et al. often underperforms compared to transformers, our method surpasses transformer benchmarks across all tested tasks.
>
> > What is the best method to evaluate the unified framework beside the interpretability analysis?
>
> First, beyond interpretability, our findings in Section 4.3 and Appendix B present empirical evaluations in non-XAI domains, such as weakly supervised semantic segmentation and in-context learning tasks. These results demonstrate the broader applicability and effectiveness of our framework.
>
> Regarding empirical evaluations, our formulation is versatile and extends to a wide range of applications, including mechanistic interpretability, regularization techniques, weakly supervised learning, and more (see 'Applications of Attention Matrices' in Section 2), making it challenging to determine which one is the best.
>
> In addition to empirical evaluations, our method provides important insights. **Our framework offers deeper insights into the internal dynamics and parametrization of modern gated-linear RNNs**. Specifically, our formulation reveals how these layers efficiently and implicitly parametrize thousands of data-dependent attention heads. These attention matrices exhibit properties such as locality and smoothness (enabled by Conv1D operations), alongside the capacity to capture sparse and complex features (via gating mechanisms and non-linear activations). In architectures like Mamba-2 and RetNet, these matrices are further refined through implicit, data-dependent normalization achieved via the LayerNorm applied at the end of the block.
>
> We will expand on these insights in a newly added discussion section in the revised manuscript.

---

> > ### Author Response · Authors · 2024-12-01
> >
> > Dear Reviewer xsWF,
> >
> > Thank you for the time and effort you've dedicated to reviewing our submission. We've done our best to address your concerns, including discussing the significance of our results and detailing the broader applicability of our formulation beyond interpretability. Additionally, based on your feedback, we have included a new appendix (see Appendix E in the revised manuscript) that elaborates on the insights derived from our formulation.
> >
> > As we near the end of the discussion period, we kindly request your engagement and feedback to ensure that all concerns are thoroughly addressed. Your insights are invaluable, and we deeply appreciate your thoughtful input.

---

### Author Response · Authors · 2024-11-16
**Official Comment by Authors**

We genuinely appreciate the reviewers' insightful and constructive feedback, which has significantly improved our manuscript and inspired further advancements in our research.

To begin, we deeply value the positive feedback provided by the reviewers. Notable examples include the following:
> This is a pretty complete paper (by Reviewer rDjF).

> It is important for the community to learn about such work (by Reviewer xsWF)

> The unified framework makes it easier to study and compare the different sequence modelling algorithms (by Reviewer xsWF).

> The paper offers a clear explanation of how implicit attention could be used to interpret gated RNNs, making it accessible to readers interested in explainability across model types (by Reviewer JKNZ).

In response to the concerns raised by reviewers JKNZ and PBPq, we conducted additional experiments and incorporated the results into the **revised manuscript** (see Table 3, with modifications highlighted in blue). **Specifically, we have included benchmarking against the recent MambaLRP** work by Jafari et al. (NeurIPS 2024) [1], which introduced an LRP-based XAI technique for Mamba models. As demonstrated in Table 3, **our method consistently and significantly outperforms this new baseline**, establishing a new SoTA for providing explanations for Mamba models. Notably, our approach achieves over a 10% improvement across all metrics, with an average gain exceeding 15%. We also note that our framework applies and was tested across multiple architectures, while MambaLRP was applied only to Mamba.

Moreover, in response to reviewer xsWF, we added a new section in Appendix E that discusses the insights arising from our formulation regarding the inner dynamics and parametrization of gated-linear RNNs. In short, our formulation reveals how these layers efficiently and implicitly parametrize thousands of data-dependent attention heads. These attention matrices exhibit properties such as locality and smoothness (enabled by Conv1D operations), alongside the capacity to capture sparse and complex features (via gating mechanisms and non-linear activations). In architectures like Mamba-2 and RetNet, these matrices are further refined through implicit, data-dependent normalization achieved via the LayerNorm applied at the end of the block.

Additional comments and responses are provided in the threads below.

Please let us know if there are any remaining questions or concerns we can address.

___
[1] MambaLRP: Explaining Selective State Space Sequence Models. Jafari et al. NeurIPS 2024

---

### Public Comment · ~Da_Xiao1 · 2024-11-27
**Thanks for your work!**

It is very insightful and helpful to me. A few questions on technical details:
1. I have some confusion on the shapes of tensors in Eq (9). E.g. why is W_x' of shape L * L? According to Eq (8), x is L * D. Shouldn't W_x'
 = diag(SILU(Linear_2(x))) be D' * L * L? (I assume Linear_2.W has shape D * D', though I'm not sure what D' should be.) And what's the shape for the final resulting tensor H? It would be very helpful if you could explicitly give shapes for each tensor in Equations 9, 11, 17, 21, 24 and 34.

2. In Appendix E you mentioned: "This implies that these components collectively enable the model to efficiently and implicitly express thousands of data-dependent (where transformers of the same size typically have around a few dozen), complex attention heads."
Why are there **thousands of** data-dependent heads in gated-linear RNN layers?

---

> ### Author Response · Authors · 2024-12-01
>
> Thank you for the thoughtful questions and kind words!
>
> 1. For clarity and simplicity, some equations in the method section omit the channel dimension, similar to how NumPy operators broadcast the first dimension. In response to your question, we will include an appendix in the next revision that extends the formulation from the method section, explicitly specifying the shapes of all tensors, including the channel dimension. Please note that these details can be easily inferred from our code (see supplementary material, lines 236-246 in MambaAccurateHiddenAttnNLP-main\mamba_ssm\ops\selective_scan_interface.py).
>
> 2. Our analysis demonstrates that **each channel** in a gated-linear RNN implicitly parametrizes a unique, complex attention matrix. This emerges from the interplay of the linear recurrent layer, activation functions, Conv1D layers, gating mechanisms, normalizations, and other architectural components. These matrices are dynamically shaped by the input data and model parameters, resulting in a distinct attention matrix for each channel.
>
>    For example, Mamba-1.3B, with 2048 channels, implicitly generates 2048 attention matrices. In contrast, transformers of similar sizes, such as Pythia-1.4B, explicitly define only 16 attention heads.

---

### Meta-Review · Area_Chair_hfKX · 2024-12-16

**Metareview:**

This paper provides a unified view of attention-free architectures as implicit causal self-attention layers, which they apply to explainability. They also show good results on semantic segmentation and in-context learning.

Overall, the reviewer's side on the acceptance. The main criticism is that the idea that these models exhibit attention-like behavior has been noted before, but the paper does a good job comparing. An interesting direction could have been exploring the tradeoffs between the complexity of the implicit attention formulation vs accuracy

**Additional Comments On Reviewer Discussion:**

The reviewers moderately engaged, maintaining their positive scores after the discussion, but no reviewer championed the paper for acceptance except PBPq.

---

### Decision · Program_Chairs · 2025-01-22

Accept (Poster)